# Uterine Artery Doppler in Complicated Twin Pregnancies

**DOI:** 10.3390/diagnostics15131696

**Published:** 2025-07-03

**Authors:** Dagmara Filipecka-Tyczka, Anna Scholz, Monika Szpotańska-Sikorska, Katarzyna Muzyka-Placzyńska, Artur Pokropek, Michał Rabijewski, Bożena Wroczyńska, Marcin Wieczorek, Małgorzata Zielińska, Magdalena Rudzińska, Krzysztof Berbeka, Paulina Pawłowska, Aleksandra Nowińska, Grzegorz Szewczyk

**Affiliations:** 1St. Sophia Hospital, 01-004 Warsaw, Poland; d.filipecka@gmail.com (D.F.-T.); zumyka@tlen.pl (K.M.-P.); wroczynska@poczta.onet.pl (B.W.); mw877@wp.pl (M.W.); go007@wp.pl (M.Z.); magdalena.rudzinska@onet.pl (M.R.); berbekakrzysztof@gmail.com (K.B.); grzegorz.szewczyk@cmkp.edu.pl (G.S.); 2First Department of Obstetrics and Gynecology, Centre of Postgraduate Medical Education, 02-097 Warsaw, Poland; 3Second Department of Obstetrics and Gynecology, Centre of Postgraduate Medical Education, 02-097 Warsaw, Poland; anna.scholz@cmkp.edu.pl; 4First Department of Obstetrics and Gynecology, Medical University of Warsaw, 02-015 Warsaw, Poland; 5Institute of Philosophy and Sociology of the Polish Academy of Sciences, 00-330 Warsaw, Poland; artur.pokropek@gmail.com; 6Department of Reproductive Health, Centre of Postgraduate Medical Education, 02-097 Warsaw, Poland; mirab@cmkp.edu.pl; 7Student Scientific Association, Department of Hygiene and Epidemiology, Medical University of Lublin, 20-059 Lublin, Poland; ppawlowska45@gmail.com (P.P.); olanowinska565@gmail.com (A.N.)

**Keywords:** uterine artery Doppler, twin pregnancies, perinatal outcome, pregnancy complications, uterine artery indices

## Abstract

**Background:** We assessed the relationship between uterine artery (UtA) indices and the occurrence of obstetrical complications in twin pregnancies. **Methods:** It was a longitudinal, prospective observation of the UtA indices and obstetric outcomes in twin pregnancies between 11 weeks of gestation and delivery. We used a logistic regression model with reliable estimators of standard errors considering the longitudinal structure. In 150 patients with twin pregnancies, 1086 ultrasound examinations were performed. The analysis incorporated nomograms for singletons and dichorionic (DC) twins. **Results:** In twin pregnancies, we observed a positive relationship between UtA indices and obstetrical complications (OR = 1.32, *p* = 0.043 for standardized PI and OR = 1.38, *p* = 0.018 for standardized RI). The risk increased with increasing UtA indices. There was a significant positive relationship between the UtA indices and analyzed pathologies in DC twins. We observed that both DC twins’ UtA indices below the 5th percentile were associated with favorable outcomes, while those above the 95th percentile were associated with adverse outcomes. According to the singleton nomograms, only the UtA PI above the 95th percentile showed significance. In MC twins, only significantly elevated UtA indices above the upper limit of nomogram were associated with adverse outcomes. **Conclusions:** The UtA nomogram for singleton and DC twins may be used in the prediction of twin pregnancy outcome, but DC nomograms are more accurate. The mechanism of obstetric complications in MC twins differs, and it requires further research. However, it seems that DC twin nomograms can be used in MC twins, but they will be less effective.

## 1. Introduction

Twin pregnancies are rare in the population. In Poland, twin pregnancies occur in 1.3% of all pregnancies, and they occur at 1.5% worldwide [1,2]. However, the twin birth rates are permanently increasing. In Poland, 21 twins per 1000 pregnancies were born in 2002, and in 2019, 26 per 1000 were born [3]. From 2002 to 2019, twin births increased by 30%.

At the same time, twin pregnancies are associated with a high risk of complications [4]. In multiple pregnancies, preterm labor (PTL), preterm premature rupture of membranes (PPROM), pregnancy hypertension (PH), pre-eclampsia (PE), gestational diabetes mellitus (GDM), fetal growth restriction (FGR), and obstetric hemorrhage are more frequent than for singletons. The risk of perinatal death increases with the number of fetuses [5,6,7]. In Polish reports, an intrauterine death entails about 0.5% of singletons and 1.4% of twin pregnancies [8].

In addition, chorionicity has an impact on the pregnancy outcome. Twenty percent of twins are monochorionic (MC), and they have additional specific complications: Twin-To-Twin Transfusion Syndrome (30% of MC twins), Twin Anemia–Polycythemia Sequence (5% of MC twins), or Twin Reverse Arterial Perfusion Sequence (1% of MC twins) [9]. In the 1950s, Ivo Brosens published a theory that the placental vessel network is correlated with pregnancy complications [10]. He analyzed the histopathological specimens of placental bed biopsies performed during cesarean section, after deliveries and miscarriages, or on specimens from perinatal hysterectomies [11,12]. He revealed that pathologies in small vessels influence the perinatal outcome. In obstetrical complications, a biopsy of the placental sites exposed a smaller diameter, thrombosis, or atherosclerosis of the spiral arteries in the basal decidua compared with uncomplicated ones [13]. That explains why abnormalities in the placental bed impact the uteroplacental blood flow. In uncomplicated pregnancies, the trophoblast infiltrates the spiral arteries, leading to thinning of the walls and widening of the vessels’ lumens. The spiral arteries pass through most of the uterine muscle and the endometrium. They provide oxygen and nutrients to the decidua, trophoblast, and placenta. They adapt to the increased demand of the growing fetus due to trophoblast invasion. As a result, vascular resistance is gradually reduced as pregnancy develops and blood flow through the uterus increases [14]. Abnormal trophoblast invasion means that its cells do not penetrate the uterine wall adequately, and the widening of the lumen of the spiral arteries is insufficient. This affects gas and chemical diffusion between mother and fetus.

According to the theory of Ivo Brosens, impaired trophoblast invasion causes all major obstetric complications, such as PE, FGR, PTL, PPROM, GDM, and intrauterine death [15,16]. The disturbance of the physiological transformation process of the uterine spiral arteries is the anatomical cause of reduced perfusion of the placenta in complicated pregnancies. Flows in the UtAs reflect the resistance and impedance of the small arteries that feed the decidua. Disturbed placentation results in abnormal flows in the UtAs. The physiological reduction in resistance does not occur [17]. Due to lateralization of the placenta, there is different remodeling and resistance in the left and right UtAs [18]. Therefore, the studies use the mean values of the flow indices for both sides. In twin pregnancies, the increased size of the placenta from two fetuses extends the trophoblast infiltration, lowers the mean resistance, and improves the impedance in the UtA compared with a singleton at the same gestational age [18]. Therefore, it is necessary to use the UtA nomograms for multiple pregnancies [19,20]. In dichorionic twin pregnancies (DC), two separate placental plates exist in one uterus.

On the other hand, in monochorionic twin pregnancies (MC), two fetuses share a single placental plate. Additionally, 90–95% of MC twins have various types of vascular anastomoses, allowing more or less balanced blood flow between them, significantly impacting the prognosis of this type of pregnancy [21,22]. Due to the different structures of the placenta, pregnancy complications and obstetric results should be analyzed separately, depending on the chorionicity [18].

UtA flows reflect the impedance of the uterine spiral arteries. Blood flow waves in UtA of the pregnancy are markers of a trophoblast invasion [15]. The development of ultrasound techniques allowed for more complex flow analyses. The UtA flow studies are prevalent in singleton pregnancies, but only a dozen reports in twin pregnancies have been published. Most clinicians analyze UtA flow to predict PE and FGR in singleton and twin pregnancies [20,23,24,25].

The research aimed to assess the relationship between UtA indices and the occurrence of obstetrical complications in twin pregnancies, both MC and DC.

## 2. Materials and Methods


**Study design**


This is a prospective single-center cohort study of UtAs blood flow in twin pregnancies designed to verify whether UtA indices change in the presence or absence of specific pregnancy complications as observed in singletons. Strobe guidelines for cohort studies were used to ensure proper presentation and reporting of results [26]. We analyzed UtA flows expressed in pulsation (PI) and resistance (RI) indices in uncomplicated and complicated twin pregnancies in individual weeks of pregnancy.


**Setting and participants**


We included patients with twin pregnancies under outpatient and inpatient care of St. Sophia Hospital in Warsaw, Poland, tertiary center, between 11 weeks of pregnancy and 0 days until the day of delivery, in the time frame from 1 February 2019 to 31 December 2020. The exclusion criteria were singleton pregnancy, triplets or more fetuses pregnancy, and childbirth outside the St. Sophia Hospital in Warsaw.

We performed 1086 ultrasound examinations in 150 twin pregnancies with a mean of 7 examinations per person. A total of 1086 PI and 1006 RI measurements were used in the final analysis: 79% of the study population were DC twins and 21% MC twins. Furthermore, 50% of DC twin pregnancies and 62% of MC pregnancies were complicated twins.

We observed the effect of gestational age on UtA indices. Therefore, we statistically corrected UtA PI and RI values for gestational age (GA).


**Variables**


In the study population, the primary outcome was the composite outcome, [27] defined as the occurrence of one or more of the following pathologies: preterm labor (PTL) before 32 weeks of gestation and 0 days, the occurrence of gestational hypertension (PH), pre-eclampsia (PE), eclampsia (E), fetal growth restriction of one or both fetuses (sFGR/FGR), preterm premature rupture of membranes (PPROM), obstetric hemorrhage, and perinatal death. The pregnancy complications were selected according to Brosen’s theory of “Great Obstetrical Syndromes” [15]. Twins were electively delivered earlier (≤38 weeks of gestation). Late and moderately premature neonates have a good prognosis (from 34 weeks and 0/7 days to 36 weeks and 6/7 days and from 32 and 0/7 to 33 weeks 6/7 days gestation, respectively) [28]. Pregnancy duration was defined based on specific twin pregnancy management. The definition of fetal growth restriction (FGR/sFGR) in twin pregnancy was based on the Delphi consensus for twins [29,30].

According to the composite outcome, we divided the study population into complicated and uncomplicated twin pregnancies. Patients who had met one of the composite outcome criteria were classified as “complicated twins”. Those that did not meet the composite outcome criteria (no PH, PE, eclampsia, FGR, sFGR, PPROM, obstetric hemorrhage, or perinatal death) were defined as “uncomplicated twins”. The latter group was additionally defined by delivery beyond 32 weeks of gestation and 0/7 days. In the manuscript, the studied pregnancies are referred to as complicated vs. uncomplicated twins.


**Chorionicity**


Dichorionic (DC) twins are pregnancies wherein each fetus has a separate placenta and separate amniotic sac; they may have the same or different sex. Monochorionic (MC) twins are pregnancies wherein two fetuses share one placenta. Monochorionic twins may have two separate amniotic sacs or may share one amniotic sac [31]. Chorionicity was defined and identified based on ultrasound criteria according to ISUOG Practice Guidelines [9]. Ultrasound criteria of DC are as follows: two separate placenta sites, different fetal sex (but same sex of fetuses does not determine chorionicity), or “lambda sing” at the attachment of membranes to one placental site in first trimester ultrasound, thick membrane (above 2 mm) separating twins. Ultrasound criteria of MC are as follows: one placental site and “T sing” at the attachment of membranes to the placenta, thin (below 2 mm) or lack of membrane separating twins in first trimester ultrasound, the same sex of twins. Twins with unknown chorionicity were excluded from the study.

“All twins” are MC and DC twin pregnancies regardless of chorionicity. “All dichorionic twins” (all DC) we defined as both complicated and uncomplicated dichorionic twins. “All monochorionic twins” (all MC) we defined as both complicated and uncomplicated monochorionic twins.


**Measurement and data sources**


We performed ultrasound examinations according to the twin pregnancy management schedule proposed by “ISUOG Practice Guidelines: role of ultrasound in twin pregnancy (2016)” [9]; MC twins at 11–14 weeks of gestation (GA), and then every two weeks from 16 GA, and DC twins at 11–14 GA, and then every four weeks from 20 GA.

We used Voluson E8 and S8 ultrasound machines (GE Healthcare Technologies, Milwaukee, WI, USA). UtA wave flows were measured prospectively using transabdominal Doppler ultrasound probes (Convex RAB6-D or C1-5 probes with a 5–10 MHz frequency with B-mode, color, and pulsed Doppler functions). The examinations were carried out strictly according to the UtA Doppler examination protocol by experienced specialists in gynecology and obstetrics. Measurements were made about 1 cm from the junction with the external iliac vessels according to the principles described by Gómez et al. [19,20,32]. We analyzed UtA indices (PI and RI) in uncomplicated and complicated twin pregnancies in individual weeks of pregnancy. In addition, the maternal, fetal, and neonatal complications and their relations with UtA indices during pregnancy were assessed. Finally, the measurements were compared with the final obstetric results (maternal, fetal, and neonatal). The data was automatically entered into the computer databases of Astraia Sofware GmbH, Ismaning, Germany Version 29.2.1 and AMMS, Asseco Poland Ltd., Cracow, Poland systems.


**Risk of bias**


To ensure the validity and reliability of our findings, we strategically addressed potential biases in our methodology. The participant selection strictly adhered to defined inclusion and exclusion criteria to mitigate selection bias. Measurement bias was minimized via standardized ultrasound protocols. The logistic regression model was adjusted for potential confounders such as maternal age, and analytical robustness was validated through sensitivity analyses.


**Statistical Methods**


The quantitative and qualitative analysis of the obtained maternal and fetal flows, fetal biometry, and amniotic fluid volume was performed. We examined the relationship between PI and RI values and obstetric complications (primary outcomes). We used a logistic regression model for the overall (unadjusted) relation between primary outcomes and standardized PI and RI (mean zero and standard deviation one across measurements separately for PI and RI). We used a linear regression model for the detailed analyses to compare levels of PI and RI (adjusted for age) between uncomplicated and complicated twin pregnancies. The dependent variables were PI or RI, and the independent was a zero–one variable, where one was the appearance of a primary outcomes, zero was a reference group with no signs, and a zero–one variable denoting chorionicity. An additional control variable was the fetal age on the examination day—naturally, the PI and RI decreased physiologically with the duration of pregnancies. Therefore, the previous study created UtA nomograms for twins [19,20,32]. To compare the results from different weeks of pregnancy, we statistically adjusted GA. The correction eliminated the effect of GA and allowed the estimation of differences. To facilitate interpretation, the results were presented as border effects (Average Adjusted Predictions [33]), i.e., the predicted values of indicators for the groups if the measurements were at the same GA. The results of “equalizing” the GA on the measurement day depend on the composition of the complicated and uncomplicated groups. Therefore, the adjusted PI and RI for the uncomplicated were different in various pathologies. The predicted values and their 90% confidence intervals were used to visualize the results [34]. We considered the results *p* < 0.05 to be statistically significant. Since UtA indices were found to differ according to chorionicity, the study population was subdivided into MC and DC pregnancies.

## 3. Results

Table 1 presents the characteristics of the study population. The groups differed in the GA at which the US examinations were performed. The mean GA was 1.5 weeks higher in all uncomplicated pregnancies than in complicated ones. If divided into subgroups, uncomplicated MC twins had a mean GA of more than 2.3 weeks higher than complicated twins, and in DC twins, it was about 1 week higher than in complicated twins (Table 2).

When we analyzed all twin pregnancies together, both PI and RI were higher in complicated twins than in uncomplicated twins. Mean PI values in complicated twins were higher by 0.083 than in uncomplicated twins, and mean RI values in complicated twins were higher by 0.029. A summary of PI and RI measurements in complicated and uncomplicated twins, with different values for MC and DC pregnancies, is presented in Table 2.

The adjusted PI analyzed for all twin pregnancies was higher for complicated than uncomplicated twins (0.852 vs. 0.803), and the difference was 0.049. It was not statistically significant (*p* = 0.145). The essential difference in the UtA indices was revealed when we split the group due to chorionicity. The adjusted PI analyzed separately behaved differently in the MC than in the DC pregnancies. In DC twins, adjusted PI was statistically higher in complicated twins than in uncomplicated twins (0.862 vs. 0.773) (Table 2). This difference was 0.098, and it was statistically significant (*p* = 0.021). On the other hand, in MC twins, the values were lower in complicated twins than in uncomplicated twins (0.860 vs. 0.832) and were not statistically significant (*p* = 0.640). A comparison of results is presented in Table 3 and Figure 1.

A similar trend was observed in the RI adjusted for GA. The adjusted RI analyzed jointly for MC and DC pregnancies was higher for complicated than uncomplicated twins (0.489 vs. 0.502). The difference was 0.013 and was not statistically significant (*p* = 0.252). In DC twin pregnancies, the RI was statistically higher for complicated twins than uncomplicated twins (0.506 vs. 0.476). This difference was 0.030 and was statistically significant (*p* = 0.020). However, in MC twins, the adjusted RI behaved differently. The correlation of UtA indices and occurrence of complications was the opposite in MC compared with DC pregnancies. The values were lower in complicated MC twins than in uncomplicated MC twins (0.494 vs. 0.513, difference = −0.019; *p* = 0.395) (Table 3 and Figure 1).

We observed a positive, statistically significant relationship between UtA PI and RI and the occurrence of obstetrical complications (composite outcome); the risk increases with increasing PI and RI in UtA (OR = 1.32, *p* = 0.043 for standardized PI and OR = 1.38, *p* = 0.018 for standardized RI) in twin pregnancies. There was a significant relationship between the UtA indices and the results for analyzed pathologies in DC twins—higher values in complicated twins vs. uncomplicated twins. However, in MC pregnancies analyzed separately, the opposite relationship occurred.

Our analysis incorporated nomograms for singletons and DC twins as established by Gómez et al. and Geipel et al. [19,32], providing a comparative backdrop for our study’s observations (Figure 2). In the assessment of MC twins using Geipel et al. nomograms, a significantly higher risk of adverse outcomes was associated with UtA PI and RI values above the 95th percentile (*p* = 0.014 and *p* = 0.003, respectively). In contrast, for the Gómez et al. singleton nomograms, no significant differences were observed in outcomes for PIs below the 5th percentile or above the 95th percentile in MC twins. Significant findings in dichorionic twins revealed that both the UtA PI and RI low and high are diagnostic. UtA indices below the 5th percentile were associated with good pregnancy outcomes at Geipel’s et al. nomograms. UtA RI above the 95th percentile at Geipel’s et al. nomograms were significantly associated with poor outcomes (*p* < 0.001). Also, the UtA PI of DC twins above the 95th percentile showed significance in the singleton nomograms of Gómez et al. (*p* = 0.001). In MC twins, only UtA indices above the 95th percentile of Geipel’s et al. nomogram were significant for prognostic poor outcomes (Table 4).

## 4. Discussion

The research aimed to assess the relationship between UtA indices and the occurrence of obstetrical complications in twin pregnancies, both MC and DC (as has been proven for singleton pregnancies). Our earlier study and those of other authors have proved that the values of UtA indices in twins are different than those in singletons, and separate normal ranges for twins have been created [19,20,32]. This relationship was investigated by looking at the adjusted levels of PI and RI in two groups. In the study, the predicted marginal effects (in fact-adjusted levels) of pulsation (PI) and resistance (RI) indices in uncomplicated and complicated twin pregnancies was compared. In the study, the adjusted PI and RI for GA were higher in complicated than uncomplicated DC twins. Statistically significant results were obtained in this group.

The volume of blood flowing in the UtAs per minute steadily increases during pregnancy [35]. This is a consequence of the decreasing impedance in the uterine spiral arteries, transformed during trophoblast invasion. Therefore, the quantification of flows in the UtA can be expressed in PI and RI, and nomograms of UtA indices both for twins and singletons decrease with the GA [19,30,32]. The EVENTS study, which analyzed the prediction of PE in twins at 11 weeks and 0 days–13 weeks and 6 days of gestation, had identical observations. The median of UtA PI for twins was lower than that for singletons in both groups—complicated and uncomplicated pregnancies (1 MoM UtA PI for twins 34 vs. 41.2 for singletons) [36]. Comparison of UtA indices’ distribution in twins and singletons in trials of Geipel et al., Filipecka-Tyczka et al., and Gómez et al. revealed lower values in twin than in singleton pregnancies [19,20,32]. We have observed a similar mechanism in twins, regardless of chorionicity or complications. We compared our results with Gómez et al. singleton UtA PI nomograms on Geipel et al. DC twins UtA PI nomograms and with Geipel et al. DC twins UtA RI nomograms. As was discussed in our previous study, UtA indices in healthy twin pregnancies are lower than those in singletons [20]. According to this data, we decided to analyze boundary values separately. In the current study, we observed that values in DC twins and all twins below the fifth percentile on Geipel’s DC twins nomograms were statistically significantly correlated with a lower incidence of the composite outcome (*p* < 0.05). Patients with low UtA PI and RI values in DC twins are likelier to have normal pregnancy outcomes. Moreover, RI in DC and MC twins above the 95th percentile on Geipel’s nomograms were statistically significantly correlated with poor pregnancy outcomes (more often meeting the composite outcome, *p* < 0.05). However, these values are in normal ranges for singletons (sic). According to all analyzed UtA nomograms, RI above the 95th percentile is correlated more frequently with poor pregnancy outcomes in twins (regardless of chorionicity). The UtA PI in MC twins above the 95th percentile in Geilpel’s twins nomogram statistically significantly correlated with a higher incidence of the composite outcome (*p* < 0.05). However, these data were not statistically significant for Gomez’s singleton nomograms, probably due to the small number of measurements (*n* = 16, *p* = 0.236). The UtA PIs in DC twins were statistically significantly correlated with incidence composite outcome only as results were above the 95th percentile for Gomez’s singletons (*p* < 0.05), so they are much higher than the 95th percentile for Geilpel’s twins. In DC twins, UtA PIs are more similar in their values to singleton pregnancies. However, the UtA indices results in MC twins are much lower. The analysis of MC pregnancies is challenging due to the small size of the groups and a deficient percentage of uncomplicated pregnancies and measurements within the normal ranges, which creates serious research problems. These data suggest established separate percentile nomograms of UtA PI and RI for MC twins.

At the same time, the obtained results confirm that lower values of UtA PI and RI correlate with a better pregnancy outcome.

The results of the study strongly correlate with Brosens’ theory of “Great Obstetrical Syndromes” [15] that pathologies of vascular networks, such as atherosclerosis or thrombosis, lead to pregnancy complications. On the other hand, PI and RI values in MC twins showed the opposite tendency, and they did not meet statistical significance. The different inclination of vascular flow rates in UtA in twin pregnancies between DC and MC may result from the different structures of the placenta. Massini et al. revealed a significant difference between mean UtAs PI in twins according to chorionicity. MC twins had a mean UtA PI of 0.07 higher than that of DCS twins (*p* < 0.05) [37]. Vascular anastomoses are present in MC twin pregnancies, which may affect the vascular impedance in these placentas. As Tian et al. and Sebire suggested, the placenta’s total size establishes the UtA’s blood flow rather than the genetic or hormonal factors related to chorionicity [38,39].

This study showed lower mean gestational age in complicated than uncomplicated twin pregnancies, which may correlate with the severity of pathologies. Higher values of PI and RI in UtA were found in complicated than in uncomplicated twins, especially after including chorionicity. There was statistical significance for DC pregnancies. These results are similar to the data for singletons. UtA indices are higher in complicated singleton pregnancies than in uncomplicated ones [40]. The UtA indices in DC pregnancies behave similarly to singletons. Physiologically, the values of pulsation and resistance indices decrease with the duration of an uncomplicated pregnancy, both in singleton [32] and twin pregnancies [19,20]. The loss of physiological adaptation of the UtA probably relates to vascular insufficiency. Interestingly, the UtA flows in MC and DC twins behaved differently. The UtA indices in MC pregnancies were statistically lower in complicated than in uncomplicated twins. Nevertheless, we noticed that the UtA indices in twin pregnancies have a similar trend as in singletons but lower mean values. This result correlates with the previous studies [19,20,32]. The most significant finding of this paper is that the UtA indices in MC twins behave inversely to those of DC twins and singletons.

Few prospective studies worldwide have assessed UtA flow in multiple pregnancies [19,20,41,42]. According to our knowledge, no similar works have been published so far. At St. Sophia Hospital in Warsaw, the twin pregnancies in 2019–2020 amounted to 1.4% of the total births, which is slightly more than in the Polish population in 2019 (1.3% of the total deliveries) [8] but slightly less than in the world (1.5%) [1,43,44]. Worldwide, twin births account for about 3% of live births and have remained relatively constant over the past 15 years compared with singleton births [1,2,43,44]. The Polish population consists mainly of Caucasian patients. The homogeneity of the study group could have had a beneficial effect on the results obtained. In the study, 79% of pregnancies were DC, and 21% were MC, corresponding to the natural chorionicity distribution in the population [45]. The study population is similar to the Polish population.

Complications are more common in twin pregnancies than in singletons. In the WHOMS study published in 2018, 15.2% of twin pregnancies in Africa, Asia, and Latin America had maternal complications, 13% of them resulted in delivery < 34 weeks GA, 51% of pregnancies were complicated by FGR of the first fetus and 55% of the second fetus, and the analyzed fetal complications concerned as many as 67% of the respondents [4]. In the studied population, 52% of complicated pregnancies were found on both the maternal and fetal sides. The pathologies concerned MC twin pregnancies more often (20 MC pregnancies [57%] vs. 59 DC pregnancies [50%]), which is slightly less frequently than reported in the literature [4]. There is a need to verify whether markers of complications in singleton pregnancies can be used for twins. One such marker is abnormal flow in the uterine arteries (especially UtA PI).

As was shown in the EVENTS trial, the screening model for PE in the first trimester of pregnancy for twins can be adopted if twin nomograms of UtA PI, mean arterial pressure, and placental growth factor (PlGF) are used [36].

### Strengths and Limitations of the Study

The rare occurrence of twin pregnancies in the population makes it challenging to collect a study group that would allow for in-depth statistical analysis. This study’s results are unique, difficult to obtain, and based on a large group of twin pregnancies analyzed together with demographic data, which allowed us to compare MC and DC twins. This study covers a large group of twin pregnancies (*n* = 150) with demographic data collected longitudinally throughout the late first, second, and third trimesters of pregnancy, and it allows for comparing MC and DC twins. We analyzed a total of 1086 ultrasound scans in comparison with the data collected in the most extensive multicenter EVENTS study, which performed 3502 Doppler studies of UtA flows [36]. In another extensive study on UtA flow charts in twins, by Geipel et al., only 990 US examinations were analyzed [19].

Gynecology and obstetrics specialists qualified in ultrasound diagnostics (over ten years of experience) made all measurements and guaranteed repeatability.

Also, in our analysis, we included RI values. The conducted study evaluates the use of UtA indices in twin pregnancies in clinical practice.

The study was conducted on a homogeneous group of Caucasian women in one center. Differences between European nations (Slavs, Germans, Balts, Roma, Celts, Greeks, and Albanians) such as height, weight, and genetic predisposition may influence the Doppler nomograms of UtA in pregnant women. In addition, there may be individualized differences in UtA flows between populations, as demonstrated before [46,47,48]. However, this can be a strength or limitation of the study. It could impact the size and homogeneity of the study group. Group homogeneity limits the extrapolation of results to other populations.

The primary outcome of the study is the composite outcome, which may result in bias [49,50]. The results of different outcomes may interfere or have opposite directions, as we have seen between MC and DC twins. In addition, it may cause confusion or overemphasized conclusions about the effect of the interventions.

Our findings indicate possible trends in scientific research. The results show that further study of these observational outcomes may improve the prediction of outcomes. The UtA flows in MC need further studies to reassess if they may be incorporated into prediction models.

For future research, it is important to test if we could adapt singleton pregnancy models for twins. Therefore, in the beginning, we decided to show whether there is a difference between complicated and uncomplicated pregnancies.

Recent studies have introduced novel approaches to define reference ranges for the uterine artery (UtA) pulsatility index using fractional polynomial modeling. This method offers greater flexibility in fitting the natural variability of UtA Doppler values across gestation compared with conventional polynomial regression. For instance, Cavoretto et al. (2023) demonstrated how serial Doppler measurements across trimesters can be modeled more accurately using fractional polynomials, thereby allowing for an improved characterization of the normality curve [51]. Including such approaches in future studies may enhance the clinical applicability of Doppler indices in pregnancy risk stratification. Perhaps this new modeling will better identify the risk group in twin pregnancies.

## 5. Conclusions

The study results indicate a relationship between uterine Doppler indices and pregnancy complications in dichorionic twins. In the composite outcome, higher values of PI and RI in UtA were observed in complicated than in uncomplicated twin pregnancies, especially in the DC twin pregnancies. The UtA changes in uncomplicated and complicated DC twin pregnancies behave similarly to singletons’ indices and probably relate to vascular insufficiency. The MoM values are higher in complicated than in uncomplicated pregnancies. Flows in UtA in MC and DC twins act differently. UtA indices are higher in complicated MC twins than in uncomplicated MC twins. This is an astonishing result.

The results suggest that UtA nomograms for singleton and DC twins can be used for the prediction of twin pregnancy outcomes, but DC nomograms are more accurate. The mechanism of obstetric complications in MC twins differs, and it requires further research. However, it seems that DC twin nomograms can be used for MC twins, but they will be less effective. The UtA flows in MC twins need further studies to create separate UtA index nomograms and reassess if they may be incorporated into prediction models.

## Figures and Tables

**Figure 1 diagnostics-15-01696-f001:**
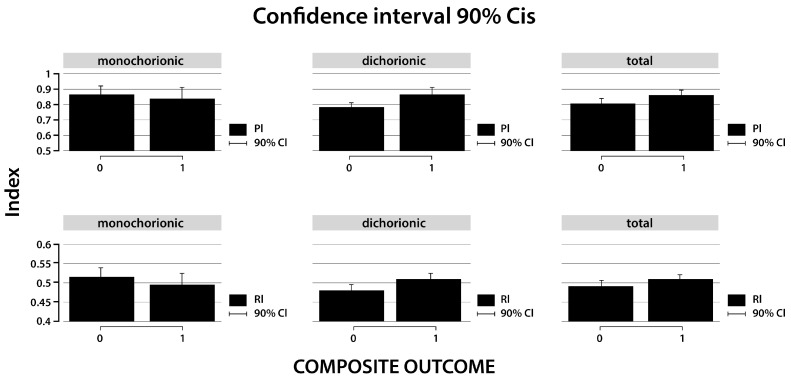
Predicted error margins of pulsation (PI) and resistance (RI) indices in uncomplicated and complicated twin pregnancies analyzed separately and in combination for chorionicity. 0—uncomplicated twins, 1—complicated twins.

**Figure 2 diagnostics-15-01696-f002:**
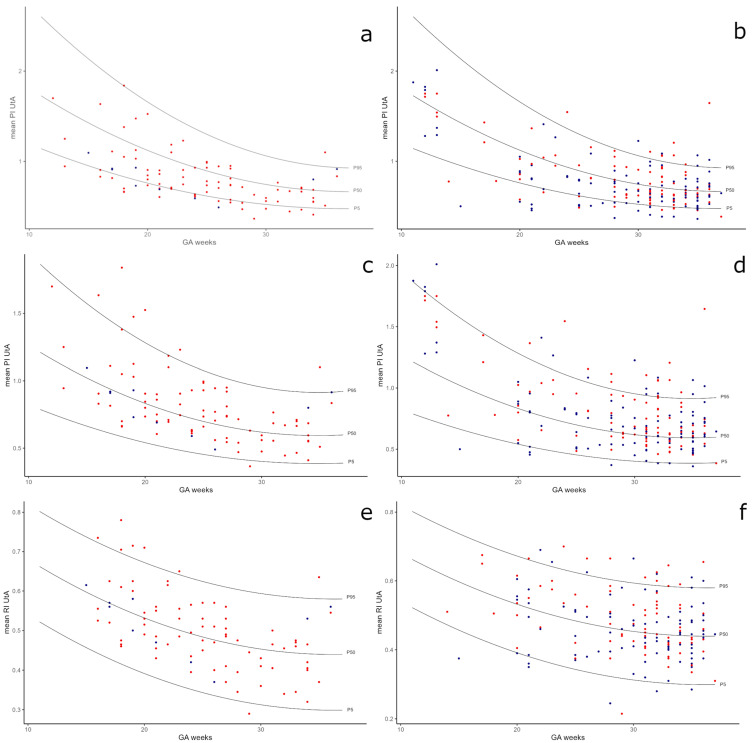
Comparison of study results to Gómez et al. singleton UtA PI nomograms (**a**,**b**), Geipel et al. DC twins UtA PI nomograms (**c**,**d**), and Geipel et al. DC twins UtA RI nomograms (**e**,**f**). Red dots—complicated twins, blue dots—uncomplicated twins.

**Table 1 diagnostics-15-01696-t001:** Characteristics of the studied population and pregnancy outcomes. The data is presented as a percentage of all pregnancies.

Chorionicity	Dichorionic	Monochorionic	Total
No. of Patients	118	32	150
Maternal age (years)	35.58 (26–38)	34.06 (27–37)	35.25 (26–38)
GA of delivery (weeks)	35.58 (26–38)	34.06 (27–37)	35.25 (26–38)
Nulliparous (n)	78 (66%)	22 (69%)	100 (67%)
BMI (kg/m^2^)	26.94 (17–44)	26.11 (20–37)	26.77 (17–44)
Race			
Caucasian (n)	118 (100%)	32 (100%)	150 (100%)
ART (n)	19 (16%)	3 (9%)	22 (15%)
Cigarette smoking (n)	5 (4%)	1 (3%)	6 (4%)
Cesarean section (n)	102 (87%)	32 (100%)	134 (89%)
PH/PE (n)	12 (10%)	3 (9%)	15 (10%)
GDM (n)	29 (25%)	4 (12%)	33 (22%)
FGR/sFGR (n)	27 (23%)	10 (31%)	37 (25%)
Sex of newborns			
Female (n)	107 (45%)	26 (41%)	133 (44%)
Male (n)	129 (55%)	38 (59%)	167 (56%)
Birth weight (mean, g)	2407.6	2190.5	2361.3
Birth weight (range, g)	(750–3460)	(820–3210)	(750–3460)
PTB (GA < 37 weeks) (n)	51 (43%)	30 (94%)	81 (54%)
PTB (GA < 32 weeks) (n)	9 (8%)	5 (16%)	14 (9%)
PPROM (n)	19 (16%)	5 (16%)	24 (16%)
Hemorrhage (n)	4 (3.4%)	1 (3.1%)	5 (3.3%)
Length of hospitalization (days)	13 (3–83)	15 (3–47)	14 (3–83)

BMI—body mass index, ART—artificial reproduction technics, PE—pre-eclampsia, PH—pregnancy hypertension, GDM—gestational diabetes mellitus, FGR—fetal growth restriction, sFGR—selective growth restriction, PTB—preterm birth, PPROM—preterm premature rupture of membranes.

**Table 2 diagnostics-15-01696-t002:** Comparison of PI and RI values in complicated and uncomplicated twin pregnancies according to chorionicity.

Composite Outcome of All Types of Chorionicity								
	variable	mean	sd	min.	max.	n	N	
Uncomplicated	PI	0.777	0.287	0.385	2.01	416	71	
	RI	0.477	0.084	0.280	0.755	385	67	
	GA	28.505	6.363	11	37	416	71	
Complicated	PI	0.860	0.365	0.365	3.560	670	79	
	RI	0.506	0.098	0.290	0.895	621	79	
	GA	26.728	6.354	11	37	670	79	
Total	PI	0.828	0.339	0.365	3.560	1086	150	
	RI	0.495	0.094	0.280	0.895	1006	146	
	GA	27.409	6.413	11	37	1086	150	
**Chorionicity**	**Monochorionic**							
	variable	mean	sd	min.	max.	n	N	MoM *
Uncomplicated	PI	0.851	0.293	0.455	1.845	94	12	1.070
	RI	0.515	0.085	0.345	0.755	90	12	1.279
	GA	27.383	5.820	12	36	94	12	
Complicated	PI	0.875	0.336	0.365	2.060	278	20	0.913
	RI	0.513	0.107	0.290	0.780	264	20	0.994
	GA	25.094	6.052	12	36	278	20	
Total	PI	0.869	0.325	0.365	2.060	372	32	0.907
	RI	0.514	0.102	0.290	0.780	354	32	0.996
	GA	25.672	6.069	12	36	372	32	
**Chorionicity**	**Dichorionic**							
	variable	mean	sd	min.	max.	n	N	MoM *
Uncomplicated	PI	0.755	0.283	0.385	2.010	322	59	1.056
	RI	0.465	0.080	0.280	0.695	295	55	0.959
	GA	28.832	6.485	11	37	322	59	
Complicated	PI	0.845	0.384	0.385	3.560	392	59	1.264
	RI	0.501	0.090	0.310	0.895	357	59	1.022
	GA	27.888	6.315	11	37	392	59	
Total	PI	0.807	0.345	0.385	3.560	714	118	1.129
	RI	0.485	0.087	0.280	0.895	652	114	1.000
	GA	28.314	6.405	11	37	714	118	

PI—pulsation index, RI—resistance index, sd—standard deviation, min—minimum, max—maximum, n—number of analyzed ultrasound examinations, N—number of patients, GA—mean gestational age at the day of ultrasound examinations, MoM—multiple of median, (*) according to normal ranges Filipecka et al. [20].

**Table 3 diagnostics-15-01696-t003:** Comparison of UtA PI and RI for complicated and uncomplicated twin pregnancies, including statistical correction for gestational age.

Status	Chorionicity	Adjusted Index	Standard Error	90% Confidence Interval		*p*-Value (*t*-Test) Difference Complicated vs. Uncomplicated
**PI**						
Uncomplicated	DC and MC	0.803	0.021	0.769	0.837	0.145
Complicated		0.852	0.026	0.810	0.895	0.145
Total	MC	0.843	0.034	0.786	0.900	0.708
	DC	0.828	0.021	0.793	0.864	0.708
Uncomplicated	MC	0.860	0.040	0.794	0.926	0.640
	DC	0.773	0.024	0.734	0.812	0.021
Complicated	MC	0.832	0.050	0.750	0.914	0.640
	DC	0.863	0.030	0.814	0.912	0.021
**RI**						
Uncomplicated	DC and MC	0.489	0.008	0.477	0.502	0.252
Complicated		0.502	0.008	0.489	0.515	0.252
Total	MC	0.501	0.012	0.481	0.521	0.673
	DC	0.495	0.007	0.483	0.506	0.673
Uncomplicated	MC	0.513	0.014	0.491	0.536	0.395
	DC	0.476	0.009	0.461	0.492	0.020
Complicated	MC	0.494	0.017	0.466	0.522	0.395
	DC	0.506	0.009	0.491	0.521	0.020

UtA—uterine artery, PI—pulsation index, RI—resistance index, MC—monochorionic twin pregnancies, DC—dichorionic twin pregnancies.

**Table 4 diagnostics-15-01696-t004:** Comparison of uterine artery pulsatility index and resistance index outcomes with Gómez et al. and Geipel et al. nomograms in monochorionic and dichorionic twin groups.

Variable	Uncomplicated		Complicated		All	*p*-Value
	N	%	N	%	N	
**All Pregnancies**						
Gomez PI < 5th percentile	176	21.67	256	18.77	432	0.100
Gomez PI > 95th percentile	36	4.43	80	5.87	116	0.151
Geipel PI < 5th percentile	48	5.91	16	1.17	64	<0.001
Geipel PI > 95th percentile	104	12.81	176	12.90	280	0.949
Geipel RI < 5th percentile	40	4.93	16	1.17	56	<0.001
Geipel RI > 95th percentile	112	13.79	248	18.18	360	0.008
**Monochorionic**						
Gomez PI < 5th percentile		28.57	160	24.54	176	0.503
Gomez PI > 95th percentile	0	0.00	16	2.45	16	0.236
Geipel PI < 5th percentile	0	0.00	8	1.23	8	0.404
Geipel PI > 95th percentile	0	0.00	64	9.82	64	0.014
Geipel RI < 5th percentile	0	0.00	8	1.23	8	0.404
Geipel RI > 95th percentile	0	0.00	88	13.50	88	0.003
**Dichorionic**						
Gomez PI < 5th percentile	160	21.16	96	13.48	256	0.632
Gomez PI > 95th percentile	36	4.76	64	8.99	100	0.001
Geipel PI < 5th percentile	48	6.35	8	1.12	56	<0.001
Geipel PI > 95th percentile	104	13.76	112	15.73	216	0.286
Geipel RI < 5th percentile	40	5.29	8	1.12	48	<0.001
Geipel RI > 95th percentile	112	14.81	160	22.47	272	<0.001

## Data Availability

All data generated or analyzed during this study are included in this article. Further enquiries can be directed to the corresponding author.

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
