# Peer review of "Uterine Artery Doppler in Complicated Twin Pregnancies"

_diagnostics, 2025, doi:10.3390/diagnostics15131696_

Round 1
Reviewer 1 Report
Comments and Suggestions for Authors
This is a well-designed and rigorously executed longitudinal prospective study assessing the relationship between uterine artery (UtA) Doppler indices and a composite of adverse obstetrical outcomes in twin pregnancies. The authors correctly stratify their analysis by chorionicity, which is fundamental to understanding the pathophysiology of twin pregnancies. The study addresses a clinically relevant question, as predicting complications in twins is a significant challenge in modern obstetrics.
The manuscript is well-written, the methodology is sound, and the statistical analysis is appropriate for the complex, longitudinal nature of the data. The conclusions are nuanced, well-supported by the results, and contribute valuable information to the field. This work stands in stark contrast to many underpowered or methodologically flawed studies on this topic and represents a significant contribution to the literature.
Nevertheless, there are a few minor points that should be addressed to improve clarity and accuracy:
- In the Methods section (line 116), the authors state: "Mean 24 examinations per person." This appears to be a significant error. With 1086 ultrasound examinations performed on 150 patients, the mean number of examinations per patient is approximately 7.2 (1086 / 150 ≈ 7.2). Please verify and correct this number, as "24" is highly implausible and could mislead the reader about the study's intensity.
- The abstract (lines 25-27) states that in DC twins, "both... UtA indices below the 5th percentile and above the 95th percentile were significantly associated with pregnancy outcomes." This is slightly ambiguous. It would be beneficial to specify this in the abstract for greater precision (e.g., "...indices below the 5th percentile were associated with favorable outcomes, while those above the 95th percentile were associated with adverse outcomes.").
- The authors use 90% confidence intervals (CIs) for their statistical analysis (e.g., Table 3, Figure 1). The standard in medical literature is 95% CIs. While using 90% is not incorrect, it is unconventional. A brief justification in the methods section for this choice would be helpful for the reader. If there is no specific reason, the authors might consider re-analyzing with 95% CIs to adhere to convention.
- Minor Typographical Errors:
There are a few minor typos that should be corrected during proofreading:
- Keywords (line 35): The keywords are numbered ("...doppler 1, twin pregnancies2..."). These numbers should be removed.
- Author Contributions (line 461): The section ends with "manuscript.d.". The final period and letter should be removed.
Author Response
Dear Reviewer,
Thank you for your insightful and constructive assessment of the text. I have made the suggested corrections.
- In the Methods section (line 116), the authors state: "Mean 24 examinations per person." This appears to be a significant error. With 1086 ultrasound examinations performed on 150 patients, the mean number of examinations per patient is approximately 7.2 (1086 / 150 ≈ 7.2). Please verify and correct this number, as "24" is highly implausible and could mislead the reader about the study's intensity.
Response 1: Changed.
2. The abstract (lines 25-27) states that in DC twins, "both... UtA indices below the 5th percentile and above the 95th percentile were significantly associated with pregnancy outcomes." This is slightly ambiguous. It would be beneficial to specify this in the abstract for greater precision (e.g., "...indices below the 5th percentile were associated with favorable outcomes, while those above the 95th percentile were associated with adverse outcomes.").
Response 2: Changed.
3. The authors use 90% confidence intervals (CIs) for their statistical analysis (e.g., Table 3, Figure 1). The standard in medical literature is 95% CIs. While using 90% is not incorrect, it is unconventional. A brief justification in the methods section for this choice would be helpful for the reader. If there is no specific reason, the authors might consider re-analyzing with 95% CIs to adhere to convention.
Response 3: Given the limited sample size and the challenges associated with data collection in this specific population, 90% confidence intervals were employed in place of the conventional 95%. This approach reflects a pragmatic trade-off between statistical confidence and interpretability, as higher confidence levels can yield excessively wide intervals in small-sample studies, potentially obscuring clinically relevant patterns. The use of 90% intervals is appropriate in exploratory analyses, where the primary objective is to detect potential trends and generate hypotheses for future research.
- Minor Typographical Errors:
There are a few minor typos that should be corrected during proofreading:
- Keywords (line 35): The keywords are numbered ("...doppler 1, twin pregnancies2..."). These numbers should be removed.
Response: Changed. - Author Contributions (line 461): The section ends with "manuscript.d.". The final period and letter should be removed.
Response 2: Changed.
Best regards,
Reviewer 2 Report
Comments and Suggestions for Authors
In this article, authors studied Doppler indices of the uterine artery in complicated/uncomplicated pregnancies with monochorionic and dichorionic twins, using nomograms for singleton pregnancies respectively dichorionic twin pregnancies. Hard work. Beautifully explained. Minor changes required:
- in Abstract you wrote:” It was a longitudinal, prospective observation of the UtA indices and obstetric outcomes in twin pregnancies between 11 weeks of gestation and delivery.” Ok.
Then, in Materials and Methods: “This is a prospective single-center cohort study of UtAs blood flow in twin pregnancies” Clear.
Later, in Materials and Methods: “We included patients with twin pregnancies under outpatient and inpatient care of St. Sophia Hospital in Warsaw, Poland, tertiary center, between 11 weeks of pregnancy and 0 days until the day of delivery, in the time frame from 02-Jan-2017 to 31-Dec-2020.” Ok.
Then, at the end of the article, you wrote:” Institutional Review Board Statement: The study was conducted according to the guidelines of the Declaration of Helsinki, and approved by the Ethics Committee of Centre of Postgraduate Medical Education in Poland (protocol code 12/BP/2019, date of approval 16/01/2019).”
Since your approval date was 2019, and patients were included starting with 2017, are you sure that this is a prospective study? Please correct it.
-in References, out of 47 titles, only 6 are recent. Out of these 6 recent ones, 2 are written by the authors (recent, good, worthy self-citation). Please replace some of the old titles with some recent ones.
Author Response
Dear Reviewer,
Thank you for your insightful and constructive assessment of the text.
Comment 1: Since your approval date was 2019, and patients were included starting with 2017, are you sure that this is a prospective study? Please correct it.
Response 1. A fair point about the dates - several people read the text before sending the manuscript, but no one noticed such an obvious typographical error.
Comment 2: In References, out of 47 titles, only 6 are recent. Out of these 6 recent ones, 2 are written by the authors (recent, good, worthy self-citation). Please replace some of the old titles with some recent ones.
Response 2: We added some current citations. Unfortunately, there are not many contemporary works on UtA analyses in twins. Hence our publication.
Best regards,